# High-Q Quasi-Bound States in the Continuum in Terahertz All-Silicon Metasurfaces

**DOI:** 10.3390/mi14101817

**Published:** 2023-09-23

**Authors:** Ruiqing Jiao, Qing Wang, Jianjun Liu, Fangzhou Shu, Guiming Pan, Xufeng Jing, Zhi Hong

**Affiliations:** Centre for THz Research, China Jiliang University, Hangzhou 310018, China; rqingjiao@163.com (R.J.); 18895364746@163.com (Q.W.); jianjun@cjlu.edu.cn (J.L.); fzshu@cjlu.edu.cn (F.S.); gmpan@cjlu.edu.cn (G.P.); jingxufeng@cjlu.edu.cn (X.J.)

**Keywords:** bound state in the continuum, guided mode resonance, all-dielectric metasurface, terahertz

## Abstract

Bound states in the continuum (BIC)-based all-silicon metasurfaces have attracted widespread attention in recent years because of their high quality (Q) factors in terahertz (THz) frequencies. Here, we propose and experimentally demonstrate an all-silicon BIC metasurface consisting of an air-hole array on a Si substrate. BICs originated from low-order TE and TM guided mode resonances (GMRs) induced by (1,0) and (1,1) Rayleigh diffraction of metagratings, which were numerically investigated. The results indicate that the GMRs and their Q-factors are easily excited and manipulated by breaking the lattice symmetry through changes in the position or radius of the air-holes, while the resonance frequencies are less sensitive to these changes. The measured Q-factor of the GMRs is as high as 490. The high-Q metasurfaces have potential applications in THz modulators, biosensors, and other photonic devices.

## 1. Introduction

High-quality (Q)-factor resonance is important for fundamental and applied research due to the enhanced light-matter interaction. In recent years, bound states in the continuum (BIC) have become a research hotspot because they possess an infinite Q-factor, even though they exist within the continuum spectra [1,2,3,4,5,6,7,8,9,10]. The ideal BIC can be easily transformed into a high-Q quasi-BIC with Fano-like curves due to the finite size of the structure, material absorption, and other external perturbations [8,11,12,13]. Owing to the ultra-high Q-factor, BIC metasurfaces have shown great potential in various applications, including high-sensitivity sensing [14,15,16,17], ultra-low threshold lasers [18,19,20,21], and nonlinear harmonic generations [22,23,24].

Thus far, significant progress has been made in high-Q THz BIC-based metasurfaces on various mechanisms, including Friedrich-Wintgen BICs (F-W BIC) [4,5,6,25], symmetry-protected BICs [26,27,28,29], and BICs originated from guided mode resonance (GMR) [30,31,32,33,34]. By breaking the symmetry of the structure or employing mode coupling, plasmonic metasurfaces can achieve BICs [5,28,29], including anisotropic BICs and chiral BICs [35,36]. However, due to the Ohmic loss of metal, the measured Q-factors are limited. The highest Q-factor reported up to now in THz metallic metasurfaces is 227 [37]. In contrast, because of the lack Ohmic loss, the electromagnetic fields of dielectric structures are effectively confined within the interior of dielectric structures. This characteristic holds great potential for achieving high-Q BIC metasurfaces [10,38]. Among them, many reports with high Q-factors come from GMR based on all-dielectric metasurface [31,32,33].

When the lattice is confined to a layer, thereby forming a periodic waveguide, an incident optical wave may undergo a GMR by coupling to a leaky eigenmode of the layer system [39,40]. Overvig et al. first explored a class of subwavelength dielectric gratings, which they called “dimerized high-contrast gratings”. In this study, they investigated how the sharp spectral features of GMRs or quasi-BICs can be manipulated by breaking the symmetry of periodic perturbations [30]. The GMR BICs are attributed to the band-folding of the metagrating, which is transformed from a single-period at the band edge into the *Γ*-point in a dual-period configuration. Shi et al. proposed and experimentally realized GMR BICs in a one-dimensional all-silicon metagrating structure at the THz band [31]. GMRs with a high Q-factor are excited by periodic perturbation, achieved by changing the position or width of metagrating ridges. The Q-factor of quasi-BIC is inversely proportional to the square of the asymmetry factor *α* [31,32,41,42]. In addition, the Q-factor of the GMRs remains infinite for symmetric metagrating when subjected to oblique incidence. However, GMRs in one-dimensional grating structures lack the diversity and flexibility compared with those in two-dimensional metagratings or metasurfaces. Recently, an all-silicon THz BIC metasurface with a 2 × 1 air-hole array was proposed. Transverse electrical (TE) and transverse magnetic (TM) GMRs were excited by changing the position of the air-holes, and the measured quasi-BICs with a Q-factor up to 860 were achieved, whereas quasi-BICs excited by changing the radii of air-holes were not demonstrated experimentally [32]. Moreover, Wang et al. proposed an all-silicon BIC metasurface with a 2 × 2 air-hole array, verified that polarization-insensitive TE and TM GMRs can be simultaneously excited by changing the radius of diagonal air-holes, and achieved a record high Q-factor of 1049 in THz metasurfaces [33]. However, the resonance frequency is sensitive to changes in the position or radius of the air-holes, which is usually not convenient for the design, fabrication, and application of the metasurfaces.

Here, we theoretically and experimentally investigate an all-silicon THz BIC metasurface with a 2 × 1 array of imperforated air-holes. Quasi-BICs originated from GMRs, can be easily excited by breaking the lattice symmetry through changes in the position or radius of the air-holes, while the resonance frequency remains stable. Moreover, the designed all-silicon metasurfaces are fabricated using a combination of photolithography and deep reactive-ion etching. The Q-factors of two TE GMRs are measured, with values as high as 490 and 386, respectively, achieved by changing the position or radius of the two air-holes.

## 2. Structure Design and Simulation Results

An all-silicon metasurface, consisting of an array of two air-holes, that we designed is shown in Figure 1a. The thickness of the metasurface is *h* = 200 μm, and the depth of air-holes is *h*_1_ = 150 μm. The periods of the unit cell in the x and y directions are *Λ*_x_ = 2*Λ* = 300 μm and *Λ*_y_ = 300 μm, respectively. Firstly, a symmetric metasurface is defined as follows: the distance between the centers of two adjacent air-holes in the x direction is *Λ* = 150 μm, and the radii of the two air-holes are equal, as shown in Figure 1b. Then, two parameters, Δ*r* and Δ*d,* are introduced to define asymmetric metasurfaces. Δ*r* represents the difference between the radii of the two air-holes in the unit cell, as shown in Figure 1c. On the other hand, Δ*d* represents the displacement of the two air-holes along the x direction, as depicted in Figure 1d.

We conducted an eigenmode analysis of the symmetric metasurface with *r* = 55 μm using the eigenfrequency solver in COMSOL Multiphysics. In the calculations, periodic boundary conditions were applied in the x and y directions of the unit cell, while a perfect matching layer (PML) was utilized in the z direction. The calculated dispersion curves are shown in Figure 2a. It has been found that there are eight eigenmodes in the frequency range of 0.33–0.49 THz, which are named TE 1, TE 2, TM 3, TM 4, TE 5, TE 6, TM 7, and TM 8. At *Γ*-point, the frequencies of the four TE modes are 0.347, 0.370, 0.445, and 0.458 THz, while the frequencies of the four TM modes are 0.377, 0.385, 0.485, and 0.490 THz, respectively. Furthermore, the Q-factors of these eigenmodes, calculated from the complex eigenfrequency (Re/2Im), are all infinite (Q > 10^9^) at the *Γ*-point and off the *Γ*-point. In particular, these eigenmodes do not exist when we calculate the dispersion curves of a metasurface with a single air-hole array (*Λ*_x_ = *Λ* = 150 μm, *Λ*_y_ = 300 μm). Therefore, these eight eigenmodes are BICs originating from GMR associated with a metagrating of period 2*Λ* in the x direction [31,32,33].

The near-field electromagnetic field distributions of the eight eigenmodes in the x-z plane at the *Γ*-point are shown in Figure 2b. Combined with their distributions in the x-y plane (not shown here), we can easily observe that TE 1, TE 2, TM 3, and TM 4 are GMRs excited by the (1,0) Rayleigh diffraction of a metagrating with a period of 2*Λ* in the x direction and propagate along the x direction. On the other hand, TE 5, TE 6, TM 7, and TM 8 are GMRs excited by the (1,1) Rayleigh diffraction of the two-dimensional metagratings in both the x and y directions and propagate along the x-y plane with an angle of 45° to the x axis. Take TE 1 and TE 2, for example, they are both GMRs of TE_0_ (the subscript represents the mode order in the z-axis). However, there is a band gap between the two modes, which corresponds to two different electromagnetic field distributions according to GMR theory [31]. Specifically, the electric near-field energy of TE 1 is concentrated in the center of four adjacent air-holes in the x and y directions, exhibiting an antisymmetric distribution along the l_1_ axis. Whereas the electric near-field distribution of TE 2 is concentrated in the center of two adjacent air-holes in the y direction, which exhibits an antisymmetric distribution along the l_2_ axis. Due to the antisymmetric field distribution of the two GMRs along axis l_1_ or l_2_, the electromagnetic energy is prevented from radiating into free space, leading to the GMR BICs having an infinite Q-factor. Therefore, when we change the position of the two air-holes in the x direction (Δ*d* ≠ 0), the antisymmetric distribution of TE 1 along the l_1_ is disrupted, causing the electromagnetic energy to radiate into free space, thus the ideal BIC transitions into a quasi-BIC. In addition, when the two air-holes have different radii (Δ*r* ≠ 0), the antisymmetric distribution of TE 2 along the l_2_ is disrupted, and the ideal BIC also collapses to a quasi-BIC.

Corresponding to TE 1 and TE 2, TM 3 and TM 4 are a pair of TM_0_ GMRs. The magnetic near-field of TM 3 exhibits an antisymmetric distribution along the l_1_ axes, whereas the magnetic near-field of TM 4 has an antisymmetric distribution along the l_2_ axis. Similarly, quasi-BIC resonances can be excited by adjusting the position (Δ*d* ≠ 0) or size (Δ*r* ≠ 0) of the two air-holes by breaking the antisymmetric distribution of the two GMRs along l_1_ or l_2_. The analysis results of TE 5, TE 6, TM 7, and TM 8 are similar to the four GMRs mentioned above. However, they belong to the higher-order GMRs. Specifically, TE 5 and TE 6 are TE_1_ GMRs, while TM 7 and TM 8 are TM_1_ GMRs, which can be clearly observed in Figure 2b.

In brief, metasurface transmissions can exhibit finite Q-factor GMRs for TE 1, TM 3, TE 5, and TM 7 when the position of the two air-holes (Δ*d* ≠ 0) is changed. Similarly, changing the size of the two air-holes (Δ*r* ≠ 0) allows for the observation of GMRs for TE 2, TM 4, TE 6, and TM 8. In addition, the four TE GMRs can be excited by y-polarized THz incidence, whereas the four TM GMRs are excited by x-polarized THz incidence. 

The above results can be confirmed by the calculated transmissions of the metasurface at different Δ*r* and Δ*d* under x- or y-polarized THz incidence, as shown in Figure 2c,d, where resonances I–VIII correspond to the eight GMRs. For the symmetric metasurface (Δ*r* = Δ*d* = 0 μm), the eight GMRs (with an infinite Q-factor) do not appear within the frequency range of 0.33–0.49 THz. However, there are two resonances, IX and X, located at 0.384 THz and 0.386 THz, excited under y- and x-polarized THz incidence, respectively. Additionally, the corresponding two eigenmodes, TE 9 and TM 10, can be found in Figure 2a,e. These two eigenmodes with a low Q-factor at *Γ*-point are exactly the same as those calculated from the metasurface with a single air-hole array, and their frequencies are highly sensitive to the period *Λ*_y_. Therefore, we can conclude that TE 9 and TM 10 are TE_1_ and TM_1_ GMRs, respectively, excited by the (0,1) Rayleigh diffraction of a metagrating in the y direction.

For the asymmetric metasurfaces with Δ*r* = 25 μm or Δ*d* = 25 μm, the eight GMRs that we considered are observed in Figure 2c,d. For example, at the y-polarized THz incidence, two GMRs, I and V, located at 0.352 THz and 0.448 THz, respectively, are excited in the metasurface when Δ*d* = 25 μm. Resonances II and VI, located at 0.360 THz and 0.452 THz, respectively, are observed in the metasurface when Δ*r* = 25 μm. Similarly, at the x-polarized terahertz incidence, two GMRs, III and VII, located at 0.385 THz and 0.480 THz, respectively, are excited for the metasurface when Δ*d* = 25 μm. Resonances IV and VIII, located at 0.363 THz and 0.457 THz, respectively, are observed for the metasurface when Δ*r* = 25 μm. Furthermore, we calculated the transmissions of the metasurface under y- and x-polarized THz incidence when Δ*d* varies from 0 to 25 μm or Δ*r* varies from 0 to 35 μm, as shown in Figure 3a–d. With the decrease in asymmetric parameters Δ*d* or Δ*r*, the bandwidth of GMRs I–VIII gradually decreases and becomes zero when Δ*r* = 0 μm or Δ*d* = 0 μm. The position of the ideal BIC is shown in blue circles in Figure 3a–d. Meanwhile, the relationships between the Q-factor of resonances I–VIII and the asymmetry factors of the metasurface *α*_d_ or *α*_r_ are calculated, as shown in Figure 3e,f, where $\alpha_{r}=∆r/r$ and $\alpha_{d}=∆d/(\Lambda -2r)$. Here, Q-factors of the resonances are also calculated from the complex eigenfrequency (Re/2Im) through eigenmode analysis. Obviously, the Q-factors are inversely proportional to the square of the asymmetry factor. However, this does not work for IV and VIII when α is relatively larger, which may be related to the saturation of the diffraction of metagratings.

## 3. Experimental Results

The designed metasurfaces were fabricated on a high-resistivity silicon wafer (>5000 Ω·cm) with a thickness of 200 μm. We chose high-resistance silicon because of its relatively low absorption loss and low dispersion in the terahertz band. The all-silicon metasurfaces with a size of 15 × 15 mm^2^ were fabricated through photolithography, followed by deep reactive ion etching (Bosch process). The specific preparation process for the sample is as follows: the silicon wafer is cleaned and spun with photoresist. Using contact photolithography, the dual air-holes structure is exposed and developed in the photoresist. This pattern is then etched through a depth of 150 μm in a 200-μm thick silicon wafer. After the plasma gas etching, the remaining photoresist is finally removed from the surface [43]. According to the structure shown in Figure 1, the parameter Δ*r* of four samples is selected as 20, 25, 30, and 35 μm, respectively, while Δ*d* of the other four samples is 10, 15, 20, and 25 μm. Figure 4e,f show the microscope images of two fabricated samples, where Δ*r* = 25 μm and Δ*d* = 25 μm, respectively.

Frequency domain spectroscopy is capable of measuring the frequency-dependent characteristics of certain materials, including the THz metasurface. It generates THz waves of a certain frequency through the difference frequency and measures the amplitude and phase of the received signal. The experimental setup in this work is built around the TeraScan 1550 frequency domain spectroscopy platform and its proprietary software manufactured by TOPTICA Photonics AG in Munich, Germany. It has two tunable 1550 nm DFB lasers (one is cooled, the other is heated in order to achieve a difference in wavelengths) and InGaAs photoconductive antennas. Frequency ranging from 0.1 to 2.9 THz can be obtained by the instrument [44].

We measured the transmission spectra of metasurfaces using a high spectral resolution (140 MHz) THz frequency domain spectral system. In the experiment, a THz beam is collimated and then focused by an off-axis parabolic lens with a numerical aperture of 0.33 onto the metasurface at normal incidence. The sample is placed on an aperture in the collimated beam, and the diameter of the spot illuminating the sample is approximately 6–8 mm. The outgoing THz waves are gathered by an off-axis parabolic lens and collimated by a collimator lens before being received by the detector. The measurements were conducted at room temperature and in a dry air condition (humidity ≤ 1%) to eliminate water vapor absorption, and the integration constant was set to 300 ms in order to enhance the signal-to-noise ratio (SNR) of the system. Figure 4a shows the calculated and measured transmission spectra of the metasurfaces with different Δ*d* when illuminated by a THz beam polarized along the y-axis. Three TE GMRs, I, V, and IX, can be clearly observed in all four samples. For example, when Δ*d* = 10 μm, the calculated resonance frequencies of I, V, and IX are 0.348, 0.445, and 0.385 THz, respectively. However, the measured resonance frequencies are slightly different, with values of 0.348, 0.434, and 0.380 THz. This slight deviation in resonance frequencies can be attributed to fabrication errors. In addition, as Δ*d* decreases from 25 to 10 μm, the resonances I and V experience very slight redshifts (<4 GHz). However, the Q-factors of the two resonances, obtained by Fano fitting, increase rapidly. Q-factors are extracted from transmissions by Fano fitting [45,46,47]:(1)$${T\left(\omega \right)=T}_{0}+A_{0}\frac{{\left[q+2\left(\omega -\omega_{0}\right)/\gamma \right]}^{2}}{1+{\left[2\left(\omega -\omega_{0}\right)/\gamma \right]}^{2}}$$
where q is the Fano fitting parameter that determines the asymmetry of the resonance curve, ω_0_ and γ represent the central resonance frequency and the resonance linewidth, respectively, T_0_ is the transmittance baseline shift, and A_0_ is the coupling coefficient. Therefore, Q = *ω*_0_/*γ*. For the four metasurfaces, the simulated Q-factors of resonance V are 370, 558, 961, and 2112, while the measured Q-factors are 124, 214, 268, and 490, respectively. This is mainly attributed to the finite size of the samples because the Q-factor of GMR is greatly related to the illumination size of the incidence. In addition, fabrication defects are also not conducive to the realization of high-Q resonance [13].

The calculated and measured transmissions of metasurfaces with different Δ*r* are shown in Figure 4b. Similarly, three TE GMRs II, VI, and IX can be observed in the measured samples, and their frequencies are less sensitive to Δ*r*. A very strong resonance VI was measured in all four samples. As Δ*r* decreases, the Q-factor of VI increases rapidly, and the highest Q-factor measured in the metasurface when Δ*r* = 20 μm is 386.

Moreover, the TM GMRs were also measured in all samples when illuminated by the x-polarized THz wave. Two transmissions of the metasurfaces with Δ*d* = 25 μm and Δ*r* = 35 μm, respectively, are shown in Figure 4c,d. We can see that the resonance X is well measured and agrees with the calculations for the metasurface with Δ*d* = 25 μm. However, the resonances III and VII that were considered are not clearly observed in the measurements. Whereas for the metasurface with Δ*r* = 35 μm, the resonance IV is not observed. A strong resonance VIII is measured with a Q-factor of 98, which agrees well with the simulation. However, it seems to be more difficult to observe QBICs of our metasurfaces under x-polarized light illumination as compared to the case of y-polarized incident light. This can be attributed to the diffraction efficiency of metagrating with a finite number of periods for different polarization incidences. The decrease in *Λ*_y_ will greatly improve the observation of TM GMRs for a certain sample size [32,33].

Finally, a comparison of the influence of Δ*d* or Δ*r* on the resonance frequency for different lattice constants *Λ*_y_ when *Λ*_x_ = 150 μm is shown in Figure 5. It is easy to see that as *Λ*_y_ increases from 150 μm to 300 μm, the influence of Δ*d* or Δ*r* on the resonance frequencies becomes weaker and weaker. For example, as Δ*d* varies from 5 μm to 25 μm, the frequency of resonance V of the metasurface with *Λ*_y_ = 300 μm only changes by 2.6 GHz, while it changes by 22.2 GHz for the metasurface with *Λ*_y_ = 150 μm. Similarly, as Δ*r* varies from 5 μm to 35 μm, the frequency of resonance VI of the metasurface with *Λ*_y_ = 300 μm only changes by 6.8 GHz, while it changes by 81.5 GHz for the metasurface with *Λ*_y_ = 150 μm. This effect can be explained as follows: when *Λ*_y_ increases, the influence of Δ*d* or Δ*r* on the equivalent refractive index of the slab waveguide becomes weaker and weaker. The working frequency is less sensitive to the regulation parameters Δ*d* or Δ*r*, which is important for the design and application of such high-Q metasurfaces.

## 4. Conclusions

In summary, we have proposed and experimentally demonstrated a high-Q all-silicon BIC metasurface consisting of an air-hole array on a Si substrate. The low-order TE and TM GMRs induced by (1,0) and (1,1) Rayleigh diffraction of a metagrating were numerically investigated in detail. The results indicate that the GMRs and their Q-factors can be easily excited and manipulated by breaking the lattice symmetry through changes in the position or radius of the two air-holes, while the frequencies of the GMRs are less sensitive to these changes. The designed all-silicon metasurfaces are fabricated by combining photolithography and deep reactive-ion etching, and the measured results obtained through THz high-resolution spectroscopy are in good agreement with the simulations. The measured Q-factors of the two TE GMRs obtained by changing the position or radius of the two air-holes are as high as 490 and 386, respectively. We anticipate that the proposed high-Q metasurface could be applied to THz modulators, biosensors, and other photonic devices.

## Figures and Tables

**Figure 1 micromachines-14-01817-f001:**
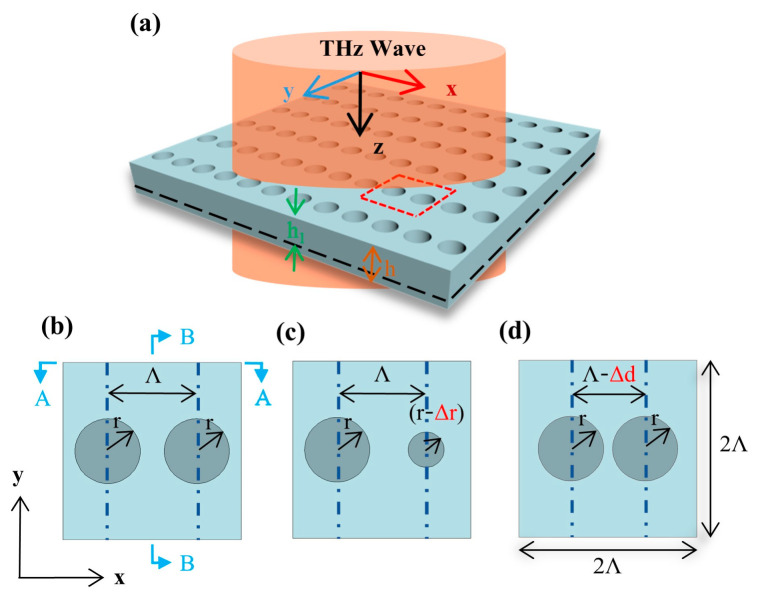
(**a**) Schematic diagram of an all-silicon metasurface consisting of two air-holes array. (**b**) Unit cell of a symmetric metasurface. The periods in the x and y directions are *Λ*_x_ = 2*Λ* and *Λ*_y_ = 2*Λ*. (**c**,**d**) The unit cell of an asymmetric metasurface is represented by Δ*r* or Δ*d*, respectively.

**Figure 2 micromachines-14-01817-f002:**
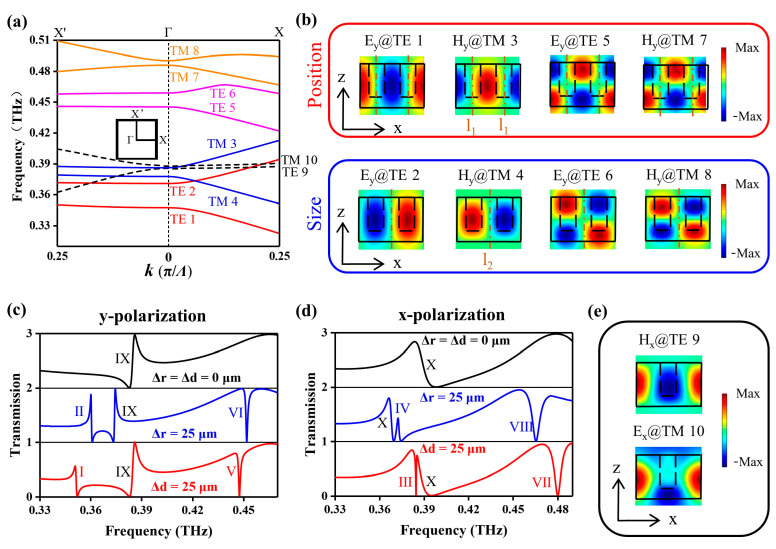
(**a**) Dispersion curves of ten related TE and TM eigenmodes for symmetrical metasurface when *r* = 55 μm, where *k* is the propagation constant, and the inset shows the first Brillouin zone of the lattice. (**b**) Near-field distribution of electric (*E*_y_) and magnetic (*H*_y_) eigenmodes in the x-z plane at *Γ*-point, in which the black dotted box represents the air-hole. Cross-sections of the near-field distributions are all taken from the A–A section in Figure 1b. (**c**,**d**) Transmissions of metasurface at different parameters Δ*r* and Δ*d* when illuminated by THz wave polarized along the y or x direction at normal incidence. The Roman letters are used to represent each resonance. (**e**) Electric (*E*_x_) and magnetic (*H*_x_) near-field distributions of two low-Q guided modes TE 9 and TM 10 at *Γ*-point in the y-z plane from B–B cross-section are shown in Figure 1b.

**Figure 3 micromachines-14-01817-f003:**
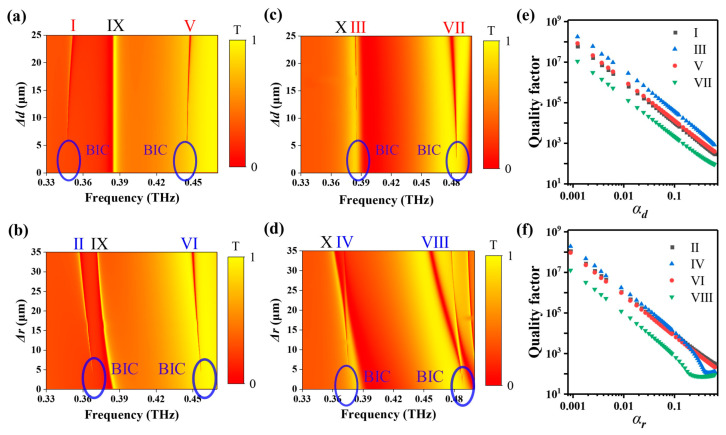
(**a**–**d**): Transmissions of the metasurface at y- and x-polarized THz incidence when Δ*d* varies from 0 to 25 μm or Δ*r* from 0 to 35 μm. (**e**) Q-factors of resonances I, III, V, and VII vs. *α_d_*. (**f**) Q-factors of resonances II, IV, VI, and VIII vs. *α_r_*.

**Figure 4 micromachines-14-01817-f004:**
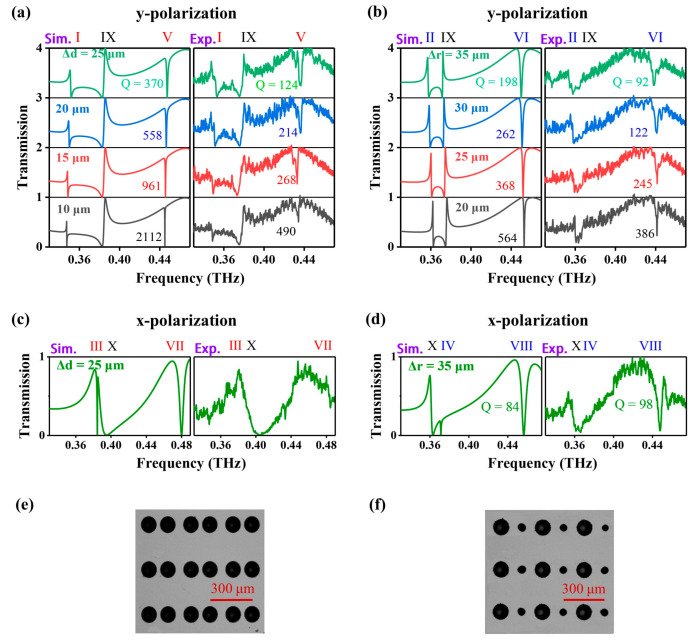
(**a**–**d**) Simulated and measured transmissions of metasurfaces under y- and x-polarized THz incidence. (**e**,**f**) Microscope images of the fabricated metasurfaces, where Δ*d* = 25 μm and Δ*r* = 25 μm, respectively.

**Figure 5 micromachines-14-01817-f005:**
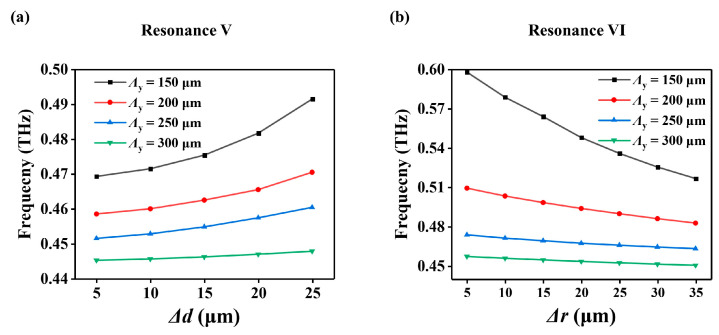
Resonance frequency with respect to Δ*d* or Δ*r* for different *Λ*y when *Λ*_x_ = 150 μm. (**a**) Resonance V. (**b**) Resonance VI.

## Data Availability

The data are available upon reasonable request from the corresponding author.

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
