# Peer review of "High-Q Quasi-Bound States in the Continuum in Terahertz All-Silicon Metasurfaces"

_micromachines, 2023, doi:10.3390/mi14101817_

Round 1

Author Response

Sep. 19, 2023

Manuscript ID: micromachines-2625036
Title: High-Q quasi-bound states in the continuum in terahertz all-silicon metasurfaces

Author: Zhi Hong; China Jiliang University

Thank you very much for giving us the chance to revise our manuscript. We would like to express our gratitude to the reviewers for taking the time to read our manuscript and provide valuable comments. We have carefully considered their feedback and incorporated it into this revised version of the manuscript. We have also taken this opportunity to recheck the entire manuscript. Based on the reviewers' comments, we have made the following changes to the manuscript.

In this manuscript, QBIC resonances in terahertz (THz) frequencies generated from all-silicon airhole metasurface are numerically and experimentally investigated. Through changes in the position or radius of the air-holes, multi-QBICs are excited under different linearly-polarized incident light.

The findings of this work may be beneficial to the design of novel QBIC devices. I would like to recommend the publication of this work after a major revision.

Response: Thanks for reviewer’s positive comments.

  1. The authors claimed that the resonance frequency is less sensitive to the changes in the position or radius of the air-holes and can remain stable. What is the corresponding mechanism to elucidate this property? Since the configuration of air-hole array is very common in the previous researches, it is better to distinctly point out the difference or novelty.

Response:

Thank the reviewer. In Fig. 5, we added more calculations of the resonance frequency vs lattice constant Λy, and explain as follows: As Λy increases from 150 μm to 300 μm, the influences of Δd or Δr on the resonance frequencies becomes weaker and weaker. This effect can be explained as that when Λy increases, the influence of Δd or Δr on the equivalent refractive index of the slab waveguide becomes weaker and weaker.

  1. According to the previous researches for QBIC, the relationship between Q factors and asymmetry parameter should satisfy the relationship: Q = 1/α2 However, the relationship in Fig. 3(f) does not work for â…£ and â…§ when α is relatively larger. Please give the corresponding reason and explain how Q factors are calculated in this manuscript.

Response:

We have revised and added in the manuscript that “Here, Q-factors of the resonances are also calculated from the complex eigenfrequency (Re/2Im) through eigenmode analysis.”

”However, this does not work for â…£ and â…§ when α is relatively larger, which may be related to the saturation of diffraction of metagratings.

  1. According to the experimental results in Figure 4, for samples with the same parameters and experimental system why it seems to be more difficult to observe QBICs under x-polarized light illumination, as compared to the case of y-polarized incident light.

Response:

Thanks. We have revised in the experimental results as It seems to be more difficult to observe QBICs of our metasurfaces under x-polarized light illumination, as compared to the case of y-polarized incident light, this can be attributed to the diffraction efficiency of metagrating with a finite number of periods for different polarization incidence. The decrease of Λy will greatly improving the observation of TM GMRs for a certain size of sample [32,33] ”

  1. The observation of TM GMRs in (Nat. Commun. 2023, 14, 2811.) is quite clear for different ΔL, which is not consistent with the description ‘…were not observed experimentally either’ in line 221. Please check whether this description is appropriate.

Response:

Thanks. We have checked the literature and find the description is inappropriate. We have corrected it in the page 7 of this manuscript.

  1. According to Fig.3(d), when Δr=35μm, â…§ is located at 0.48 THz, not 0.45 THz. Therefore, the observed resonance in Fig.4 (d) is not ‘VIII’.

Response:

We have carefully checked the electromagnetic field distribution of the resonance â…§ and adjacent resonance at Δr = 0 μm and 35 μm, and we are sure that the resonance â…§ is located at 0.45 THz when Δr = 35μm.

  1. The description of experimental setup should be adequate.

Response:

We have added the description of experimental setup in page 5 of this manuscript.

  1. The full name of GMR in the abstract should be given when it firstly occurs.

Response:

We have corrected it in the abstract.

In addition to the aforementioned revisions, we have also reviewed the format of the manuscript, including the references, and made the necessary changes to ensure that the manuscript meets the style requirements for Micromachines.

Sincerely,

Zhi Hong

Reviewer 2 Report

Dear Editor,

the article of R. Jiao, Q Wang et al., titled 'High-Q quasi bound states in the continuum in terahertz all-silicon metasurfaces' presents simulations and experiments of metamaterials for manipulation of light at terahertz frequencies.

In more detail, the authors design a metamaterial that sustains bright,radiative and dark (nonradiative) electromagnetic modes. By breaking the symmetry, some of the dark modes start to radiate and take the form of quasi bound states in the continuum. The slightest the symmetry break, the higher their quality factor.

The article is well-written, the research is appropriately designed, the results look convincing and interesting. 

I recommend to accept this work and I only have few comments for the Authors in case it goes to revision.

1) You fit the experimental results with Fano resonances, which are described as a resonance interacting with a continuous background. Maybe you can give more emphasis in the Fano interference and explain what is the oscillator and the background here. 

2) Maybe it is better to give the fitted formula (Fano +any background) - as slightly different definitions exist - and some details on the fitting since it is important for extracting the width, intensity and quality factors from the experimental data. 

3) To improve the introduction, it could be helpful to have a sketch illustrating bound states in the continuum versus other modes.

Author Response

Sep. 19, 2023

Manuscript ID: micromachines-2625036
Title: High-Q quasi-bound states in the continuum in terahertz all-silicon metasurfaces

Author: Zhi Hong; China Jiliang University

Thank you very much for giving us the chance to revise our manuscript. We would like to express our gratitude to the reviewers for taking the time to read our manuscript and provide valuable comments. We have carefully considered their feedback and incorporated it into this revised version of the manuscript. We have also taken this opportunity to recheck the entire manuscript. Based on the reviewers' comments, we have made the following changes to the manuscript.

the article of R. Jiao, Q Wang et al., titled 'High-Q quasi bound states in the continuum in terahertz all-silicon metasurfaces' presents simulations and experiments of metamaterials for manipulation of light at terahertz frequencies.

In more detail, the authors design a metamaterial that sustains bright,radiative and dark (nonradiative) electromagnetic modes. By breaking the symmetry, some of the dark modes start to radiate and take the form of quasi bound states in the continuum. The slightest the symmetry break, the higher their quality factor.

The article is well-written, the research is appropriately designed, the results look convincing and interesting.

I recommend to accept this work and I only have few comments for the Authors in case it goes to revision.

Response: Thanks for reviewer’s positive comments.

1) You fit the experimental results with Fano resonances, which are described as a resonance interacting with a continuous background. Maybe you can give more emphasis in the Fano interference and explain what is the oscillator and the background here.

Response: 

Thank the reviewer. Like other articles on BIC, quasi-BICs in this manuscript exhibit asymmetric curves, thus, we just use of Fano fitting to extract Q-factors, but do not analysis its origin, maybe from interference of GMR and F-P background. 

2) Maybe it is better to give the fitted formula (Fano +any background) - as slightly different definitions exist - and some details on the fitting since it is important for extracting the width, intensity and quality factors from the experimental data.

Response:

Thanks. We have added the Fano fitting formula in page 6 of the manuscript. 

  • To improve the introduction, it could be helpful to have a sketch illustrating bound states in the continuum versus other modes.

Response: 

The most notable feature of BIC that distinguishes from other modes in metasurface is its high Q-factor. In introduction, we have added more comparisons of high-Q resonance in plasmonic BIC metasurfaces with dielectric BIC metasurfaces, and the highest Q factors achieved experimentally are 227 in plasmonic BICs, 1049 in dielectric BIC metasurfaces.

In addition to the aforementioned revisions, we have also reviewed the format of the manuscript, including the references, and made the necessary changes to ensure that the manuscript meets the style requirements for Micromachines.

Sincerely,

Zhi Hong

Reviewer 3 Report

Comments on the manuscript

The work discusses the development and experimental demonstration of high-quality (Q) all-silicon metasurfaces that utilize Bound States in the Continuum (BIC) in the terahertz (THz) frequency range. The metasurface consists of an array of imperforated air-holes on a silicon substrate. The research explores how low-order TE and TM Guided Mode Resonances (GMRs) are induced by Rayleigh diffraction in metagratings, particularly when the lattice symmetry is disrupted by changes in air-hole position or radius. The study reveals that the GMRs have high Q-factors and are easily manipulated. The metasurfaces have potential applications in THz modulators, biosensors, and other photonic devices.

In my assessment, the manuscript is interesting, and the theoretical/experimental findings contribute to BIC metasurfaces, a topic of high interest. Overall, the manuscript is methodologically robust, with well-supported claims and conclusions. Consequently, this manuscript aligns with the scope of Micromachines, pending the resolution of a few minor concerns. Please find my suggestions and comments for the authors below.

To begin with, this paper initially creates a negative impression as it lacks a thorough perspective in the first paragraph of the introduction section. For instance, it primarily focuses on dielectric BIC studies, while completely overlooking the recent advancements in plasmonic BICs. It is recommended to incorporate more comprehensive examples of recent progress in the realm of plasmonics, including: (1) anisotropic plasmonic BICs ["Bound states in the continuum in anisotropic plasmonic metasurfaces." Nano Letters 20.9 (2020): 6351-6356]; (2) the intriguing topic of chiral BICs in plasmonic metasurfaces.

Quoting “The ideal BIC can be easily trans-27 formed into a quasi-BIC, exhibiting a high Q-factor Fano resonance, due to the finite size of the structure, material absorption, and other external perturbations” It's important to note that Fano resonances were not studied by Fano himself in terms of a new kind of resonance. Therefore, referring to them as "Fano resonances" may not be entirely accurate from a historical perspective. A more appropriate terminology might be "high-Q quasi-BICs with Fano-like curves" to emphasize the resemblance to Fano profiles without implying a direct historical connection to Fano's work. This revised phrasing would maintain precision while acknowledging the resemblance to Fano curves. Please comment.

“We can easily observe that TE 1, TE 2, TM 3, and TM 4 are 108 GMRs excited by the (1,0) Rayleigh diffraction of a metagrating with a period of 2Λ in the x direction. On the other hand, TE 5, TE 6, TM 7, and TM 8 are GMRs excited by the (1,1) Rayleigh diffraction of the two-dimensional metagratings in both the x and y directions.” This claim needs to be supported by evidence. I can provide you with hints: Given that each supercell has a period of 300 μm, and the entire structure is positioned on a silicon substrate, you can determine at which wavelength the Rayleigh diffraction orders (±1,0) exclusively take place at normal incidence based on empty lattice equations.

Fitures 4a and 4b. What led to the observed experimental Q-factors being several times lower than what was predicted through simulation?

readable

Author Response

Sep. 19, 2023

Manuscript ID: micromachines-2625036
Title: High-Q quasi-bound states in the continuum in terahertz all-silicon metasurfaces

Author: Zhi Hong; China Jiliang University

Thank you very much for giving us the chance to revise our manuscript. We would like to express our gratitude to the reviewers for taking the time to read our manuscript and provide valuable comments. We have carefully considered their feedback and incorporated it into this revised version of the manuscript. We have also taken this opportunity to recheck the entire manuscript. Based on the reviewers' comments, we have made the following changes to the manuscript.

The work discusses the development and experimental demonstration of high-quality (Q) all-silicon metasurfaces that utilize Bound States in the Continuum (BIC) in the terahertz (THz) frequency range. The metasurface consists of an array of imperforated air-holes on a silicon substrate. The research explores how low-order TE and TM Guided Mode Resonances (GMRs) are induced by Rayleigh diffraction in metagratings, particularly when the lattice symmetry is disrupted by changes in air-hole position or radius. The study reveals that the GMRs have high Q-factors and are easily manipulated. The metasurfaces have potential applications in THz modulators, biosensors, and other photonic devices.

In my assessment, the manuscript is interesting, and the theoretical/experimental findings contribute to BIC metasurfaces, a topic of high interest. Overall, the manuscript is methodologically robust, with well-supported claims and conclusions. Consequently, this manuscript aligns with the scope of Micromachines, pending the resolution of a few minor concerns. Please find my suggestions and comments for the authors below.

Response: Thanks for reviewer’s positive comments.

  1. To begin with, this paper initially creates a negative impression as it lacks a thorough perspective in the first paragraph of the introduction section. For instance, it primarily focuses on dielectric BIC studies, while completely overlooking the recent advancements in plasmonic BICs. It is recommended to incorporate more comprehensive examples of recent progress in the realm of plasmonics, including: (1) anisotropic plasmonic BICs ["Bound states in the continuum in anisotropic plasmonic metasurfaces." Nano Letters 20.9 (2020): 6351-6356]; (2) the intriguing topic of chiral BICs in plasmonic metasurfaces.

Response:

Thanks. We have added the description of BICs in plasmonic metasurfaces in the introduction of this manuscript.

  1. Quoting “The ideal BIC can be easily transformed into a quasi-BIC, exhibiting a high Q-factor Fano resonance, due to the finite size of the structure, material absorption, and other external perturbations” It's important to note that Fano resonances were not studied by Fano himself in terms of a new kind of resonance. Therefore, referring to them as "Fano resonances" may not be entirely accurate from a historical perspective. A more appropriate terminology might be "high-Q quasi-BICs with Fano-like curves" to emphasize the resemblance to Fano profiles without implying a direct historical connection to Fano's work. This revised phrasing would maintain precision while acknowledging the resemblance to Fano curves. Please comment.

Response:

Thanks. We have corrected "Fano resonances" to "resonances with Fano-like curves" in the introduction of this manuscript.

  1. “We can easily observe that TE 1, TE 2, TM 3, and TM 4 are GMRs excited by the (1,0) Rayleigh diffraction of a metagrating with a period of 2Λ in the x direction. On the other hand, TE 5, TE 6, TM 7, and TM 8 are GMRs excited by the (1,1) Rayleigh diffraction of the two-dimensional metagratings in both the x and y directions.” This claim needs to be supported by evidence. I can provide you with hints: Given that each supercell has a period of 300 μm, and the entire structure is positioned on a silicon substrate, you can determine at which wavelength the Rayleigh diffraction orders (±1,0) exclusively take place at normal incidence based on empty lattice equations.

Response:

Thanks for reviewer’s good suggestions. Actually, resonances IX and X also come from the Rayleigh diffraction orders (±1,0) by metagrating with a period of 300 μm. In fact, from electric and magnetic near-field distributions in x-z plane and x-y plane, it is easy to judge the GMRs induced by the Rayleigh diffraction orders (±1,0) or (1,1). We have revised it in the manuscript.

  1. Figures 4a and 4b. What led to the observed experimental Q-factors being several times lower than what was predicted through simulation?

Response:

Thanks. We have added the explanation for this as “This is mainly attributed to the finite size of the samples, because the Q-factor of GMR is greatly related to the illumination size of the incidence. In addition, fabrication defects are also not conducive to the realization of high-Q resonance.”

In addition to the aforementioned revisions, we have also reviewed the format of the manuscript, including the references, and made the necessary changes to ensure that the manuscript meets the style requirements for Micromachines.

Sincerely,

Zhi Hong

Round 2

Reviewer 1 Report

The authors point to point response to my comments about the details of this study and carefully revised manuscript correspondingly. I recommend that the manuscript be accepted.